# A Quantitative Detection Method for Surface Cracks on Slab Track Based on Infrared Thermography

**Xuan-Yu Ye [1], Yan-Yun Luo [1], Zai-Wei Li [2,\*] and Xiao-Zhou Liu [3,\*]**

1   Institute of Railway and Urban Mass Transit, Tongji University, Shanghai 201804, China; yexuanyu2180199@tongji.edu.cn (X.-Y.Y.); yanyunluo@tongji.edu.cn (Y.-Y.L.)
2   School of Urban Rail Transportation, Shanghai University of Engineering Science, Shanghai 201620, China
3   College of Urban Transportation and Logistics, Shenzhen Technology University, Shenzhen 518118, China
\*   Correspondence: zaiweili@sues.edu.cn (Z.-W.L.); liuxiaozhou@sztu.edu.cn (X.-Z.L.)

**Abstract:** Surface cracks are typical defects in high-speed rail (HSR) slab tracks, which can cause structural deterioration and reduce the service reliability of the track system. However, the question of how to effectively detect and quantify the surface cracks remains unsolved at present. In this paper, a novel crack-detection method based on infrared thermography is adopted to quantify surface cracks on rail-track slabs. In this method, the thermogram of a track slab acquired by an infrared camera is first processed with the non-subsampled contourlet transform (NSCT)-based image-enhancement algorithm, and the crack is located via an edge-detection algorithm. Next, to quantitatively detect the surface crack, a pixel-locating method is proposed, whereby the crack width, length, and area can be obtained. Lastly, the detection accuracy of the proposed method at different temperatures is verified against a laboratory test, in which a scale model of the slab is poured and a temperature-controlled cabinet is used to control the temperature-change process. The results show that the proposed method can effectively enhance the edge details of the surface cracks in the image and that the crack area can be effectively extracted; the accuracy of the quantification of the crack width can reach 99%, whilst the accuracy of the quantification of the crack length and area is 85%, which essentially meets the requirements of HSR-slab-track inspection. This research could open the possibility of the application of IRT-based track slab inspection in HSR operations to enhance the efficiency of defect detection.

**Keywords:** slab track; infrared thermography; surface-crack detection; image enhancement; scale-model test



## 1. Introduction

Slab tracks are widely used in newly built high-speed railways (HSRs) due to its significant advantages, such as high stability and low maintenance cost. However, under the joint effect of the large wheel–rail force induced by high-speed trains and the environmental factors in operation, slab tracks are prone to structural defects, such as surface cracks [1,2]. To prevent these cracks, the design cracking joint (DCJ), as a structural optimization measure, is adopted in some prefabricated concrete slabs [3,4]. However, according to the operation and maintenance practice in Chinese HSR, surface cracks can still occur in the non-DCJ areas on slabs [5]. More seriously, full cracks may develop, which is a hidden danger to the operational safety of HSRs. Although the maintenance specification of the Chinese HSR formulates standards for the inspection and maintenance of surface cracks, the timely detection of these cracks is challenging. This is mainly because the current methods of crack detection mainly depend on inefficient manual inspection within a narrow time window. Therefore, it is of great practical significance to develop an efficient surface-crack-detection method for HSR slab tracks.

The current detection methods for surface cracks can be classified into two categories: contact methods and non-contact methods. The contact methods, as traditional inspection methods, usually involve the placement of sensors on the track-slab surface to detect

cracks. The main examples of these methods are the impact-echo method [6,7], the acoustic emission method [8,9], and the optical-fiber-sensing method [10,11]. These methods can accurately locate cracks and identify their sizes, but their detection efficiency is low and the requirement for operators is high. Therefore, these methods are applicable to the detection of key parts of slab tracks rather than the inspection of a whole line. In comparison, the non-contact methods are relatively new, based on the development of modern sensing technology. Characterized by high detection efficiency, these methods are increasingly applied in HSRs. In particular, the most efficient method is the use of the high-speed track-inspection train, whereby the service condition of the track structure, including some typical defects, can be detected at high speeds [12]. However, the inspection train cannot directly detect cracks on the surfaces of track slabs [13]. To solve this problem, some novel track-inspection trolleys equipped with ground-penetrating radar (GPR) [14], machine vision (with visible cameras) [15,16], and infrared thermography (IRT) [5,17,18] devices have been developed in recent years. However, because of the influence of the steel bars inside the reinforced track slab, the accuracy of the geological radar-imaging method is low. For the machine-vision-based method, the dark condition during the window time at night can significantly limit its application in HSRs. In comparison, infrared thermography, which is used to measure the temperature fields of track slabs and does not need additional illuminating devices, is more suitable for detecting surface cracks in track slabs. The existing studies are mainly exploratory discussions about the feasibility of identifying surface cracks using aging methods. There is still a lack of studies on how to effectively extract the features of slab-surface cracks and assess these cracks in a quantitative manner.

Therefore, this paper proposes a novel image-processing method, which can locate and quantify crack areas in track slabs using infrared thermography. Figure 1 illustrates the flowchart of the proposed method. The effectiveness of the proposed method at different temperatures is validated against a laboratory test, in which the temperature field of a scale model of a track slab with artificial cracks is studied. The remaining parts of this paper are as follows. The second part presents the image-enhancement and edge-detection methods in the processing of the infrared-thermography data. In the third part, an algorithm for quantitatively detecting the crack area based on the number of graphic pixels is presented. The fourth part analyzes the accuracy and effectiveness of the proposed algorithm by comparing the results of the laboratory test with the scaled model of the track slab. The last part gives the conclusions.

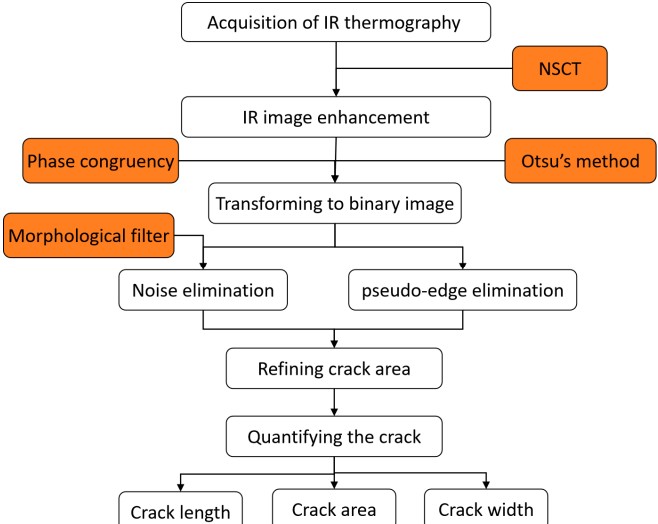

**Figure 1.** Flowchart of the proposed crack-detection method.

## 2. Infrared-Thermogram Processing

### 2.1. Acquisition of the Infrared Thermography

In this research, the Fluke IR camera (type: TIX620) was used to record the thermal image of slab surface. The key parameters of the camera are listed in Table 1. The experimental setup was presented in our previous research [5], as shown in Figure 2. The IR camera with a standard lens of 30 mm is installed on a metal bracket and the distance between the camera and slab surface is 0.75 m. The emissivity of the slab surface is 0.92.

**Table 1.** Key specifications of the IR camera (Fluke TIX620).

| Category | Value |
| --- | --- |
| Image resolution | 640 × 480 (307,200 pixels) |
| Instantaneous field of view (IFOV) | 0.85 mRad |
| Field of view (FOV) w/standard 30 mm lens | 32.7° × 24.0° |
| Frame rate (@ max. image resolution) | 30 and 9 Hz |
| Thermal sensitivity (NETD) | <0.04 °C at 30 °C target temp (40 mK) |
| Measurement range | −40 °C to 600 °C (−40 °F to 1112 °F) |
| Measurement accuracy | ±2 K or ±2% |
| Photo | |

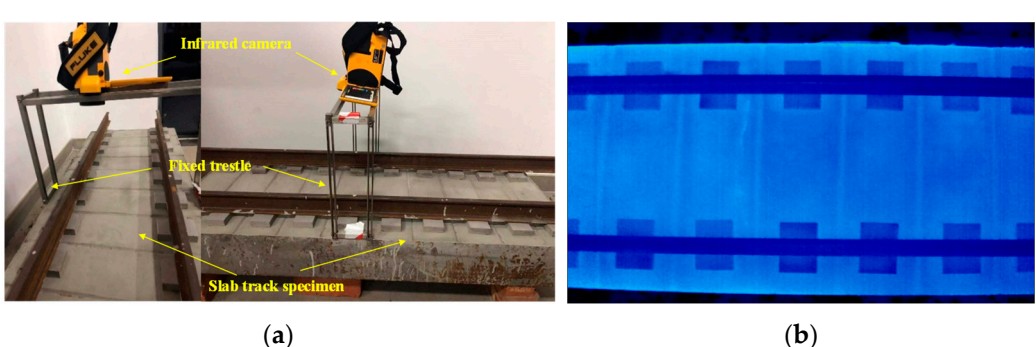

(**a**)                                                        (**b**)

**Figure 2.** Acquisition of the infrared thermography of slab track by IR camera: (**a**) experimental setup; (**b**) infrared thermography of slab track [5].

### 2.2. Non-Subsampled Contourlet Transform (NSCT)-Based Image-Enhancement Algorithm

Key to the quantitative assessment of the surface cracks in track slabs is the quality of the infrared thermography of the rail track. In this regard, this section presents the image-enhancement and edge-detection methods used to process the original infrared thermography.

The non-subsampled contourlet transform (NSCT)-based image-enhancement algorithm is used to transform the infrared thermogram from time domain to frequency domain, and modifies the frequency-domain coefficients at different levels. It can highlight the crack edges and details in the thermal image of the track-slab surface. The NSCT is characterized by multiple scales and directions, compared with traditional methods, such as wavelet transform, which are based on the scale correlation of transformation coefficients. It is composed of non-subsampled pyramid (NSP) and non-subsampled directional filter banks (NSDFB) [19]. During image decomposition, NSP decomposes the raw image into high-frequency and low-frequency components, and then NSDFB decomposes the high-frequency component into multiple band-pass directional sub-bands. The multi-scale and multi-directional decomposition principle of NSCT is shown in Figure 3.

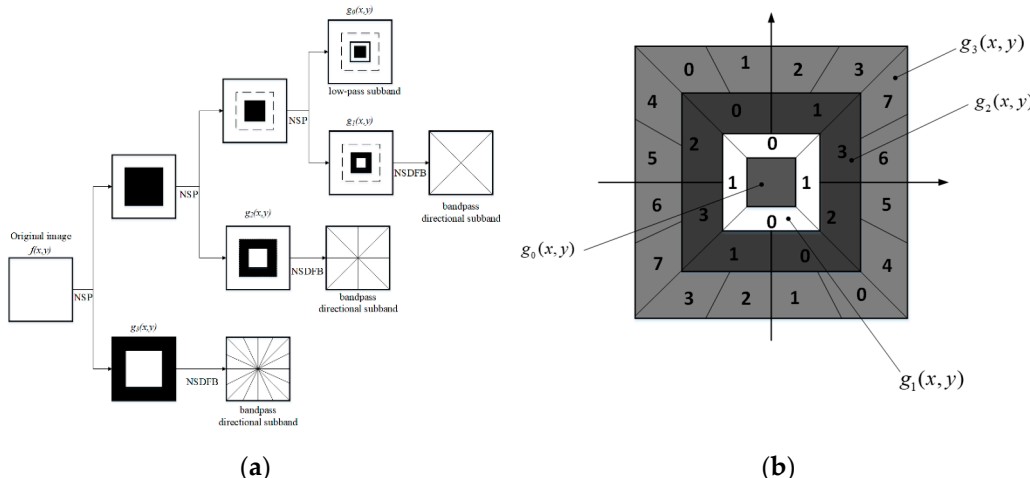

(a)                                                              (b)

**Figure 3.** Principle of NSCT transform: (**a**) decomposition structure; (**b**) sub-bands on frequency plane.

In the image-enhancement algorithm for infrared thermograms, it is necessary to select the appropriate threshold according to the transformation coefficient obtained from the original image after frequency-domain transformation. The threshold is used to distinguish the noise and effective edge details in the image, to avoid image distortion and the amplification of noise. In this paper, the threshold value is determined by the transformation coefficients at different scales and directions obtained from the original image after NSCT. The relationship between the threshold value and the standard deviation of the transformation coefficient is expressed as:

$$T_k^l = \frac{1}{2} \sqrt{\frac{1}{MN} \sum_{m=1}^{M} \sum_{n=1}^{N} \left( W_k^l f(m,n) - mean_{W_k^l} \right)^2} \tag{1}$$

where $T_k^l$ is the threshold value at the $k$-th directional sub-band of the $l$-th scale, $W_k^l f(m,n)$ is the transformation coefficient at the $k$-th directional sub-band of the $l$-th scale at coordinates $(m,n)$, $mean_{W_k^l}$ is the mean value of all transformation coefficients in the $k$-th directional sub-band on the $l$-th scale, $M \times N$ is the size of the sub-band image.

By using NSCT, the original image can be transformed to obtain transformation coefficients of multiple scales and directions. Next, the nonlinear enhancement function is adopted to modify the coefficients in the transformed domain. The enhancement function can be expressed as [19]:

$$f(x) = a[sigm(c(x-b)) - sigm(-c(x+b))] \tag{2}$$

$$a = \frac{1}{sigm(c(1-b)) - sigm(-c(1+b))} \tag{3}$$

$$sigm(x) = \frac{1}{1+e^{-x}} \tag{4}$$

where coefficient $b$ controls the enhancement range, $0 < b < 1$, $c$ controls the enhancement intensity, and its variation range is (20, 50).

To avoid the influence of the grayscale of the original image on the coefficients $a$, $b$, and $c$ in the enhancement function, the maximum transformation coefficient $x_{max}$ in the directional sub-band is used as the normalization factor to normalize the transformation coefficient.

It can be seen from Equations (2)–(4) that the domain of $f(x)$ is [–1, 1], and $f(0) = 0$, $f(1) = 1$. The coefficient $b$ in Equation (2) is determined by solving the nonlinear equation $f(x) = x$, and the enhancement function becomes:

$$f(x) = ax_{max}\left[sigm\left(c\left(\frac{x}{x_{max}} - b\right)\right) - sigm\left(-c\left(\frac{x}{x_{max}} + b\right)\right)\right] \qquad (5)$$

The thresholds at different scales and in different directions are selected in Equation (1) by using the transformation coefficients. However, if these thresholds are directly used to modify the transformation coefficients at a certain scale or in a certain direction without considering the correlation of the transformation coefficients at different scales, some key transformation coefficients may be considered as noise and set to zero. To consider the correlation between transformation coefficients at different scales, a multiscale product is introduced into the infrared-image-enhancement algorithm in this paper, which is expressed as:

$$P_k^l f(m,n) = W_k^l f(m,n) \cdot W_k^{l+1} f(m,n) \qquad (6)$$

where $P_k^l f(m,n)$ is the multiscale product of the $k$-th directional sub-band on the $l$-th scale at coordinates $(m,n)$, $W_k^l f(m,n)$ is the transformation coefficient of the $k$-th directional sub-band on the $l$-th scale at coordinates $(m,n)$, and $W_k^{l+1} f(m,n)$ is the transformation coefficient of the $k$-th directional sub-band on the ($l$+1)th scale at coordinates $(m,n)$.

It can be seen from Equation (6) that the multiscale product is the product of the transformation coefficients of the image in the same directional sub-bands of two adjacent scales. In this paper, the threshold is used to modify the multiscale product rather than the transformation coefficients to avoid the aforementioned problem. Thus, the infrared-image-enhancement algorithm can preserve more details of the original image while strengthening the crack edges and reducing the noise in the image.

The process of the proposed image-enhancement algorithm based on multiscale product is as follows:

Step 1. Use NSCT to decompose the original image to layer $L$, and obtain the transformation coefficients in sub-bands of $i$ scales and $2^i$ directions.

Step 2. Determine the threshold values for sub-bands of different scales and directions according to Equation (1), and obtain the adaptive enhancement function $f(x)$ according to Equation (5).

Step 3. Use Equation (6) to calculate the multiscale product $P_r^j f(m,n)$ of sub-bands in each direction, where $j \in [1, i-1]$, $r \in [1, 2^i]$. According to Equation (7), the threshold $T_r^j$ is directly substituted for the multiscale product $P_r^j f(m,n)$ to reduce the noise.

$$\overline{W}_r^j f(m,n) = \begin{cases} W_r^j f(m,n), & | \quad P_r^j f(m,n) \geq T_r^j \\ 0, & | \quad P_r^j f(m,n) < T_r^j \end{cases} \qquad (7)$$

where $W_r^j f(m,n)$ is the transformation coefficient of the $r$-th directional sub-band on the $j$-th scale at coordinates $(m,n)$, $P_r^j f(m,n)$ is the multiscale product of the $i$-th directional sub-band on the $j$-th scale at coordinates (m, n), $T_r^j$ is the threshold value of the $r$-th directional sub-band on the $j$-th scale, and $\overline{W}_r^j f(m,n)$ is the new transformation coefficient at the coordinates $(m,n)$ of the $r$-th directional sub-band on the $j$-th scale obtained after thresholding.

Step 4. Substitute $\overline{W}_k^l f(m,n)$ into the enhancement function $f(x)$ to conduct image enhancement.

Step 5. Conduct inverse NSCT using the transformation coefficients of all scales and sub-bands in all directions after the enhancement processing to reconstruct the infrared image of the slab's surface cracks.

### 2.3. Edge Detection of Infrared Thermogram

The current edge-detection methods for infrared thermograms are mainly based on the gray gradient of the image or the measurement standards on the gray space, so the effect of detection depends heavily on the grayscale and contrast of the image. However, in the infrared thermogram of the track slab with surface cracks, the difference between the grayscale at the crack edge and intact part is small and difficult to detect. To solve this problem, this paper adopts a crack-edge-detection algorithm based on phase congruency of the Fourier components of the infrared image of the slab-surface cracks, and sets the phase congruency with large values in the image as the feature points of the crack edge. Through these steps, the crack area in tan infrared image can be detected. Since the phase congruency is not affected by the gray gradient at the crack edge in the infrared image, it has a higher detection accuracy than the edge-detection algorithm based on gray gradient and gray space.

The Fourier expansion of one-dimensional signal $t(z)$ can be obtained by Fourier transformation:

$$t(z) = \sum_d A_d \cos(d\omega z + \varphi_{d0}) = \sum_d A_d \cos(\varphi_d(z)) \tag{8}$$

where $A_d$, $\varphi_{d0}$, $\omega$ are the amplitude, initial phase, and angular frequency of the $d$-th Fourier component of the signal $t(z)$, respectively. The $\varphi_d(z)$ is the phase value of the $d$-th Fourier component at $z$.

According to the previous research on the phase characteristics of Fourier series of various signals [20], phase congruency can be defined as:

$$PC(z) = \max_{\overline{\varphi}(z) \in [0,2\pi]} \frac{\sum_d \cos[\varphi_d(z) - \overline{\varphi}(z)]}{\sum_d A_d} \tag{9}$$

where $\overline{\varphi}(z)$ is the weighted average phase value of the Fourier components of the signal $t(z)$ when $PC(z)$ takes the maximum value at point $z$. It can be seen from Equation (9) that in order to maximize the phase congruency $PC(z)$, the sum of the differences between the phase $\varphi_d(z)$ and $\overline{\varphi}(z)$ of each Fourier component of signal $t(z)$ must reach the minimum value. To solve problem of computation cost in Equation (9), Venkatesh et al. [21] introduced the concept of local energy. The relationship between local energy function, phase−congruency function, and Fourier components of the signal is shown in Figure 4.

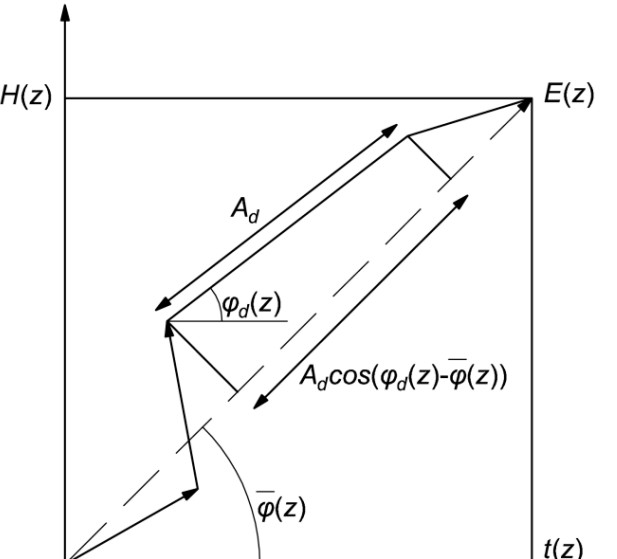

**Figure 4.** Relationship between local energy function, phase−congruency function, and Fourier components of signal [21].

It can be seen from Figure 4 that the local energy can be expressed as:

$$E(z) = \sqrt{t^2(z) + H^2(z)} \tag{10}$$

where $H(z)$ is the Hilbert transform of $t(z)$. It can also be seen from Figure 4 that the local energy $E(z)$ is the sum of the projections of the Fourier components of the signal $t(z)$ in the $E(z)$ direction:

$$E(z) = \sum_d A_d \cos(\varphi_d(z) - \overline{\varphi}(z)) \tag{11}$$

The combination of Equations (9) and (11) yields:

$$PC(z) = \frac{E(z)}{\sum_d A_d} \tag{12}$$

Equation (12) shows that the phase congruency is the ratio between the local energy and the sum of the amplitudes of the Fourier components of the signal, and Equation (11) shows that the local energy is closely related to the phase of the Fourier components. It can be seen that the phase congruency is greater when the phase-angle values of the Fourier components of the signal are centrally distributed.

The principle of crack-edge detection is based on the fact that the distribution of the phase-angle value of each Fourier component at the crack edge is relatively concentrated, which means the phase congruency in the cracks is higher than those in the non-crack areas. The process of edge-detection algorithm of track-slab-surface cracks is as follows:

Step 1. Calculate the phase congruency of all pixels in the infrared thermogram of slab surface cracks.

Step 2. Multiply the phase congruency values of all pixels by 255, i.e., convert the phase−congruency values of each pixel into the corresponding grayscales, and obtain the phase−congruency image, as shown in Figure 5b.

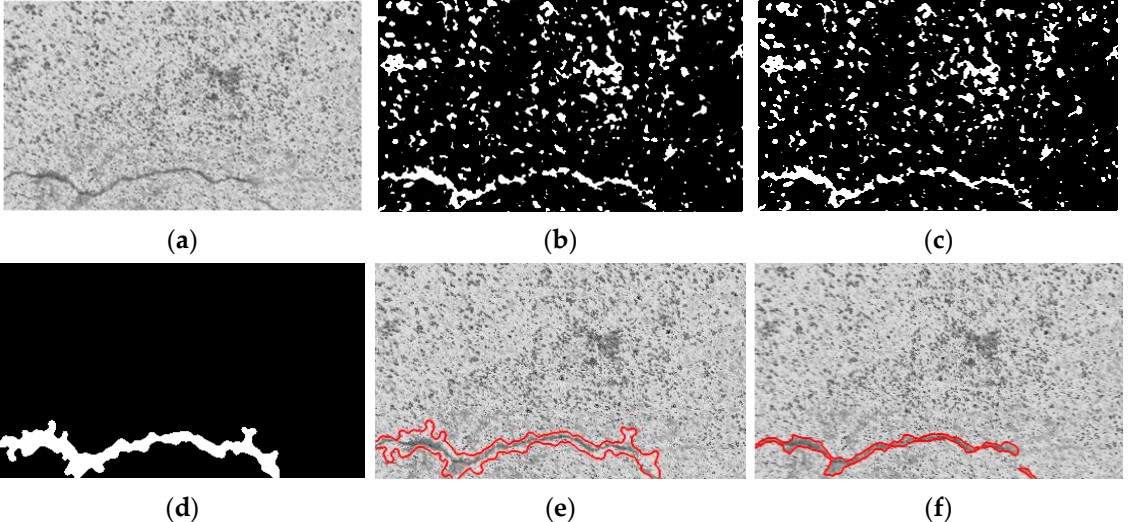

**Figure 5.** Crack-edge-detection process of infrared image of track-slab-surface crack: (**a**) Infrared image of slab surface crack; (**b**) phase congruency image; (**c**) binary image after segmentation; (**d**) processed binary image; (**e**) crack area (before refining); (**f**) refined crack area.

Step 3. Use Otsu's method to solve the segmentation threshold of the phase−congruency image of the slab-surface crack, obtain the area with a large gray value (i.e., phase−congruency value), and convert the segmented phase−congruency image into a binary image, as shown in Figure 5c.

Step 4. Use the morphological filter to solve the over-segmentation and under-segmentation problems caused by Otsu's method. In the morphological filter, the dilation

operation is used to connect the fracture points in the crack area and the open operation is used to eliminate the isolated noise points and maximumly preserve the structure of the crack area.

Furthermore, it should be noted that the track-slab surface is not smooth and features defilement, and there are temperature-distortion points in the collected infrared images, which can generate blocky pseudo-edge areas in the segmented images. By comparing the distribution of the pseudo-edge area and the crack area in the segmented images, it is found that the number of pixels occupied by the pseudo-edge area due to the defilement and temperature distortion is lower than the number of pixels in the crack area. Furthermore, the crack appears as a long and narrow strip area in the segmented image, while the pseudo-edge area appears as a block area, which means the length–width ratio of the crack area is greater than that of the pseudo-edge area.

In this regard, in this paper, the crack area and pseudo-edge area are marked in the binary image after morphological processing as connected domain $\{bw_1, bw_2, \cdots, bw_{sum}\}$, where $sum$ is the total number of crack areas and pseudo-edge areas. Next, the total number of pixels, and the lengths of the major axis and the minor axis of all connected domains are calculated. The major axis and the minor axis refer to the major and minor axis of the ellipse with the same standard second-order central moment as the area. Next, the total number of pixels, as well as the ratio of the length of the major axis to the minor axis of all connected domains, are sorted, and two sets $V_1$, $V_2$ are formed, accordingly, by thresholding the number of pixels. The threshold value of $V_1$ is set to $thv_1$, i.e., the pseudo-edge area can be defined as the connected domain with the number of pixels less than $thv_1$; the threshold value of $V_2$ is set to $thv_2$ and $thv_2 = \alpha * max\{V_2\}$, where $\alpha$ is an empirical parameter and $\alpha \in [0.4, 0.7]$, i.e., the pseudo-edge area can be defined as the connected domain with a ratio of the length of the major axis to that of minor axis less than $thv_2$. After morphological processing and elimination of pseudo-edge areas, a full image of the crack area can be extracted from the segmented binary image, as shown in Figure 5d.

Step 5. Refine the crack area. It can be seen from Figure 5e that the crack area obtained in Step 4 is larger than the actual crack area in the infrared thermography, and the redundant part is the slab concrete near the crack edge, so the crack area obtained in the previous step needs to be refined.

Because the thermal conductivity of the air in the surface cracks is quite different from that of the concrete material, heat accumulates more easily inside the cracks, and the speed of heat dissipation is also slower than that in the non-crack areas, resulting in differences in temperature values and temperature gradients between the cracks and non-cracks on the slab surface. Since the thermal imager can convert the temperature value of the slab surface by detecting the infrared radiation emitted from the slab surface, a temperature-value matrix, which contains the temperature value of all pixels in the infrared thermogram, can also be obtained.

The temperature-value matrix of the infrared thermogram is defined as:

$$\text{TEM} = \begin{bmatrix} tem_{1,1} & \cdots & tem_{1,col} \\ \vdots & \ddots & \vdots \\ tem_{row,1} & \cdots & tem_{row,col} \end{bmatrix} \tag{13}$$

where $tem_{1,col}$ is the temperature value of the pixel at the coordinate $(1, col)$ and $row \times col$ is the size of the image.

The transverse temperature-gradient matrix of infrared thermogram is defined as:

$$gradx = \begin{bmatrix} gdx_{1,1} & \cdots & gdx_{1,col-1} \\ \vdots & \ddots & \vdots \\ gdx_{row,1} & \cdots & gdx_{row,col-1} \end{bmatrix} \tag{14}$$

where $gdx_{row,col-1} = tem_{row,col} - tem_{row,col-1}$.

The vertical temperature-gradient matrix of infrared thermogram is defined as:

$$grady = \begin{bmatrix} gdy_{1,1} & \cdots & gdy_{1,col} \\ \vdots & \ddots & \vdots \\ gdy_{row-1,1} & \cdots & gdy_{row-1,col} \end{bmatrix} \tag{15}$$

where $gdy_{row-1,col} = tem_{row,col} - tem_{row-1,col}$.

According to the pixel coordinates in the crack area obtained in Step 4, three scalar attributes are given to each pixel in the area, temperature value, transverse temperature gradient value, and vertical temperature-gradient value, so that a dataset with three-dimensional scalar attributes $\{AT_1, AT_2, \cdots\}$ is formed. The "$k$-means" clustering algorithm [22] is used to classify all elements in the dataset, and the crack area is refined according to the classification results and the coordinates of the elements in the dataset. The effect of crack-edge extraction is shown in Figure 5f. Note that the category number $clu$ is set to 4 after several trials.

In Figure 5, the whole process of the crack-edge-detection algorithm based on phase congruency is presented. It can be observed that a complete binary image of cracks from the infrared thermogram of the track slab can be extracted, which can be used for subsequent measurement and analysis of surface cracks.

## 3. Quantitative Detection Method of Crack Area

To detect the surface crack in a quantitative manner, the pixel width, length, and total number of pixels in the connected area of the infrared thermogram are calculated and the pixel calibration is conducted to obtain the width, length, and area of the crack.

### 3.1. Pixel Length of the Crack

Since the slab-surface cracks appear uneven in the thickness direction, Zhang–Suen thinning method [23] is used to refine the extracted crack area into a crack skeleton composed of single-layer pixel points, as shown in Figure 6. The pixel length of the crack can then be calculated. Based on the skeleton graph, the pixel points on the crack skeleton are coded by 8-direction chain code proposed by Freeman [24]. The chain code is a method to describe a curve by using the coordinates of the starting point of the curve and the direction of the boundary point [24]. The coding process starts from the endpoint of the crack-skeleton map, and the pixel points in the crack skeleton are encoded according to the direction in Figure 7 until all the pixels on the skeleton have been scanned.

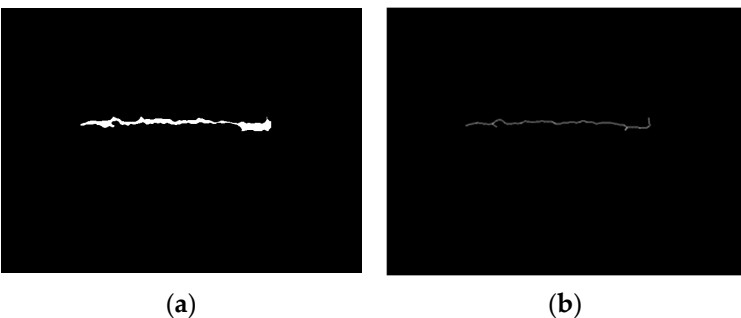

(**a**)                              (**b**)

**Figure 6.** Crack-edge-detection process of infrared image of track-slab-surface crack. (**a**) Extracted crack area. (**b**) Crack-skeleton graph.

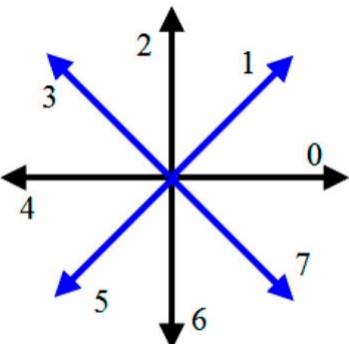

**Figure 7.** 8-direction Freeman chain code [24].

According to the 8-direction chain code of pixels on the crack-skeleton graph, the length of the crack skeleton, i.e., the pixel length of the crack, can be obtained as:

$$Len_{crack} = wei_{even}Num_{even} + wei_{odd}Num_{odd} \qquad (16)$$

where $Num_{even}$ and $Num_{odd}$ are the numbers of even chain codes and odd chain codes in pixel direction coding of the crack-skeleton image, respectively. In Figure 6, the direction of the even chain code is vertical or horizontal, and the segment length of the crack at the pixel point is the number of pixels, with a weight of $wei_{even} = 1$. The crack direction in the odd chain code is $\pm45°$, and the segment length of the crack at the pixel point is $\sqrt{2}$ times the number of pixels, with a weight of $wei_{odd} = \sqrt{2}$.

### 3.2. Pixel Area of the Crack

By counting the total number of pixel points in the crack area extracted from the infrared thermal image of the slab surface, the pixel area of the crack $Area_{crack}$ can be obtained.

### 3.3. Pixel Width of the Crack

With the pixel length and area of the crack obtained in (1) and (2), the average pixel width $\overline{width}_{crack}$ of the crack area can be calculated as:

$$\overline{width}_{crack} = \frac{Area_{crack}}{Len_{crack}} \qquad (17)$$

The local pixel width of the crack can be calculated by the minimum-distance method, which involves the extraction of the crack area from the thermogram to obtain a single-layer edge curve for the crack, as shown in Figure 8. The upper crack-edge pixel in each column in the crack-edge area is taken as the center, and the minimum distance between the lower crack-edge pixel in the adjacent $n_{arround}$ columns, which is defined as the pixel width of the column, can be calculated. Note that $n_{arround}$ is generally taken to be between 3 and 5.

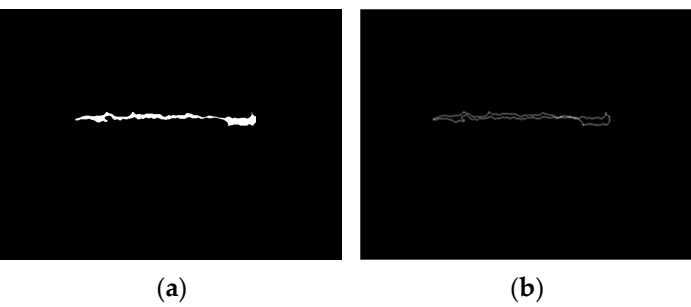

| (a) | (b) |

**Figure 8.** Edge detection of crack area: (**a**) extracted crack area; (**b**) edge of crack area.

*3.4. Pixel Calibration in Crack Region*

Another important problem that needs to be solved is the false detection of DCJ, which is the designed component of the slab track, rather than the surface crack, as shown in Figure 9. Because the DCJ is more consistent and the boundary distinction is more obvious, the false joint area can probably be detected as the surface crack on the slab track. To distinguish DCJ from actual surface crack, this paper proposes an identification method for DCJ. Since the spacing between adjacent DCJs on the track slab is a fixed value, the pixel spacing between the centroids of adjacent connected domains at the DCJ is a fixed value $H_{real}$. Therefore, the pixel distance $H_{pixel}$ between adjacent connected areas at the DCJ is obtained by calculating the centroid coordinates of the connected areas at the two DCJs. The conversion coefficient $K_{pixel}$ between a single pixel and the physical length in the infrared thermal image is obtained by comparing the actual distance $H_{real}$ and the pixel distance $H_{pixel}$ between adjacent DCJs in the infrared thermogram:

$$K_{pixel} = \frac{H_{real}}{H_{pixel}} \tag{18}$$

where $H_{real}$ is the actual distance between adjacent DCJs, $H_{pixel}$ is the pixel distance between adjacent DCJs, and $K_{pixel}$ is the physical distance of a single pixel. The conversion factor $K_{pixel}$ can be used to convert the pixel length, local pixel width, average pixel width, and pixel area of the crack area into the actual crack length, actual crack width, and actual crack area.

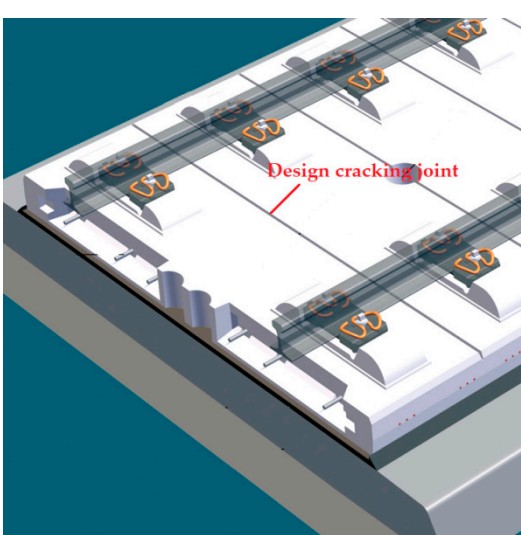

**Figure 9.** Slab-track structure and design cracking joint (DCJ).

**4. Case Study**

*4.1. Model Test*

To verify the effectiveness of the proposed algorithm, we conducted a laboratory test, in which a slab-track model was used. Considering that the actual size of the slab track (6.45 m long, 2.95 m wide along the concrete base, and 0.53 m high) was large, a scaled model of the track slab which could reflect the real temperature field of the slab under different conditions was used.

(1)    Size of the scaled model

To determine the size of the scaled model, we used finite element (FE) simulation, though which the temperature field of the track slab was simulated. The relevant environmental parameters, boundary conditions, and model verification are given in [5]. A crack with a width of 0.2 mm and a depth of 20 mm was added to the slab surface, as shown in Figure 10.

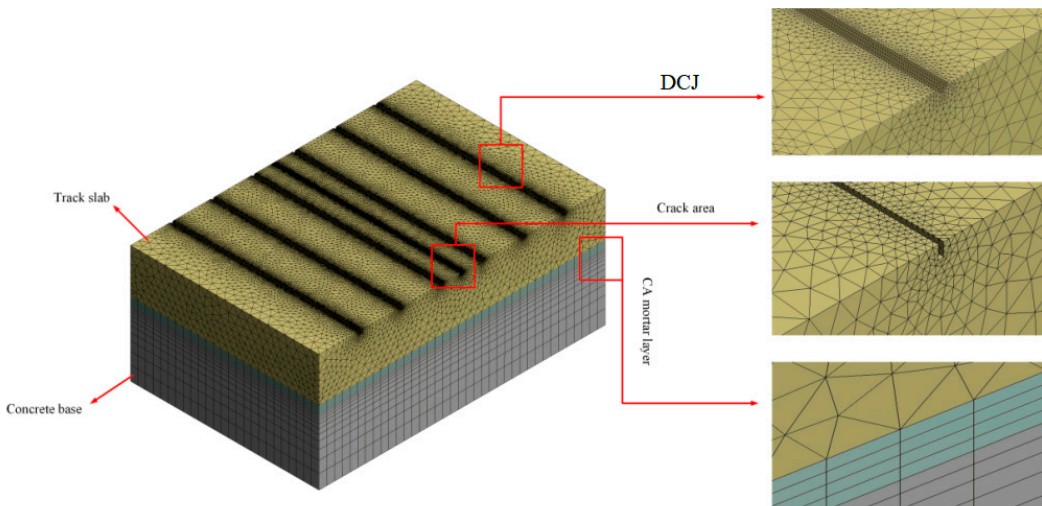

**Figure 10.** Diagram of structural model of slab track with surface crack.

For the length of the track model, the track slab was a longitudinal concrete structure, and the size of the track slab in the length direction was much larger than that in the height and width direction. Furthermore previous studies [25,26] also ignored the heat transfer in the longitudinal direction when studying the temperature field of slab-track structures and simplified it into a two-dimensional heat-transfer structure. In this regard, to determine the length of the track model, in this study, we performed a large number of trial calculations. Based on the results of the trial calculations and [27], the length of the track model was set to 1.272 m.

After the determination of the model length, the influence of the height and width on the temperature field of the track slab was further analyzed. In this study, the 1:3, 1:4, and 1:5 scale models (in both height and width direction) were chosen in the temperature calculation, and the other input parameters towered ran from [5]. The temperature values in the crack and non-crack areas on the track-slab surface at different times are shown in Figure 11.

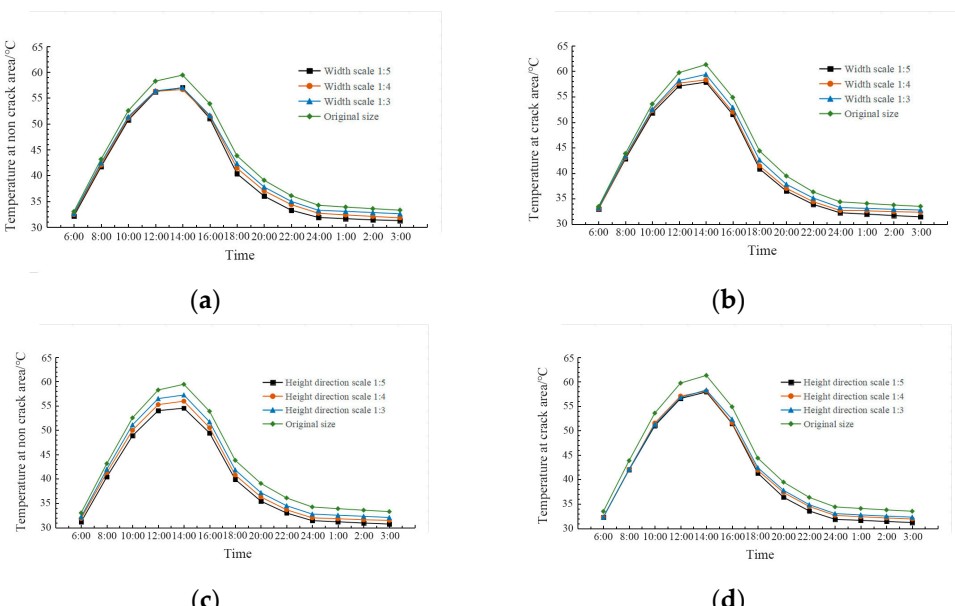

**Figure 11.** Temperature in crack and non-crack areas on the surface of track slab: (**a**) in non-crack areas with different width scales; (**b**) in crack areas with different width scales; (**c**) in non-crack areas with different height scales; (**d**) in crack areas with different height scales.

It can be seen from Figure 11 that when the reduction ratio of the scaled model in the width direction or height direction was larger (i.e., 1:3 scale), the temperature in the crack and non-crack areas on the slab surface had smaller errors with the full-scale model; the error caused by the height reduction is greater than that caused by the width direction. Therefore, the scale in the width direction was set to 1:3, i.e., 0.85 m; the height of the model was set to 0.53 m, the same size as the full-scale model.

Table 2 lists the calculation results for the temperature with the scale model and the full-scale model, as well as the error of the scale model (compared with that of the full-scale model). It was found that the maximum error of the temperature in the scale model in the non-crack areas was 2.12%, and the average error was 1.41%; the maximum error in the scale model in the crack areas was 2.26%, and the average error was 1.46%. Therefore, it can be concluded that the scale model can effectively show the surface-temperature field of an actual slab track, and it can be used to validate the thermal imaging detection of surface cracks.

**Table 2.** Calculation results of slab-surface temperature using scale and full-scale models.

| Time | Scale Model | | | | Full-Scale Model | |
|---|---|---|---|---|---|---|
| | Temperature in Non-Crack Areas/°C | Error | Temperature in Crack Areas/°C | Error | Temperature in Non-Crack Areas/°C | Temperature in Crack Areas/°C |
| 6:00 | 33.00 | 0.07% | 33.48 | 0.04% | 26.71 | 26.59 |
| 8:00 | 42.96 | 0.45% | 43.66 | 0.48% | 35.91 | 33.39 |
| 10:00 | 52.23 | 0.59% | 53.06 | 1.00% | 47.3 | 41.13 |
| 12:00 | 57.70 | 0.96% | 59.11 | 1.04% | 55.23 | 46.16 |
| 14:00 | 58.87 | 0.94% | 60.45 | 1.38% | 56.53 | 46.7 |
| 16:00 | 52.96 | 1.73% | 53.77 | 2.03% | 51.32 | 42.55 |
| 18:00 | 43.02 | 1.82% | 43.39 | 2.26% | 41.14 | 35.46 |
| 20:00 | 38.25 | 2.12% | 38.59 | 2.24% | 36.4 | 33.01 |
| 22:00 | 35.34 | 2.07% | 35.56 | 2.19% | 34.24 | 32.01 |
| 24:00 | 33.54 | 2.04% | 33.66 | 2.19% | 32.88 | 31.28 |
| 1:00 | 33.23 | 1.95% | 33.49 | 1.75% | 32.32 | 30.97 |
| 2:00 | 32.96 | 1.85% | 33.32 | 1.39% | 31.83 | 30.69 |
| 3:00 | 32.71 | 1.75% | 33.18 | 1.02% | 31.39 | 30.43 |

(2)    Track model and relevant experimental equipment

In this test, the scale model of the track slab was produced according to the parameters specified by the China High-Speed Railway. The materials of the track slab, bearing layer, and CA mortar layer and their corresponding coordination are listed in Table 3. Figure 12 shows the scale model after curing.

**Table 3.** Composition materials and mixture ratio of track slab, concrete base, and CA mortar layer (Unit: kg $\times$ m$^{-3}$).

| Ingredients | Track Slab | Concrete Base | CA Mortar |
|---|---|---|---|
| | Concrete Grade: C55 | Concrete Grade: C15 | |
| Cement | 432 | 140 | 506 |
| Fly ash | 48 | 60 | - |
| Emulsified asphalt | - | - | 265 |
| Sand | 717 | 830 | 984 |
| Gravel | 1121 | 1244 | - |
| Aluminium powder | - | - | 0.06 |
| Water | 125 | 125 | 158 |
| Water reducer | 7.2 | 1 | 5.5 |
| Expander | - | - | 86 |
| Defoamer | - | - | 0.5 |

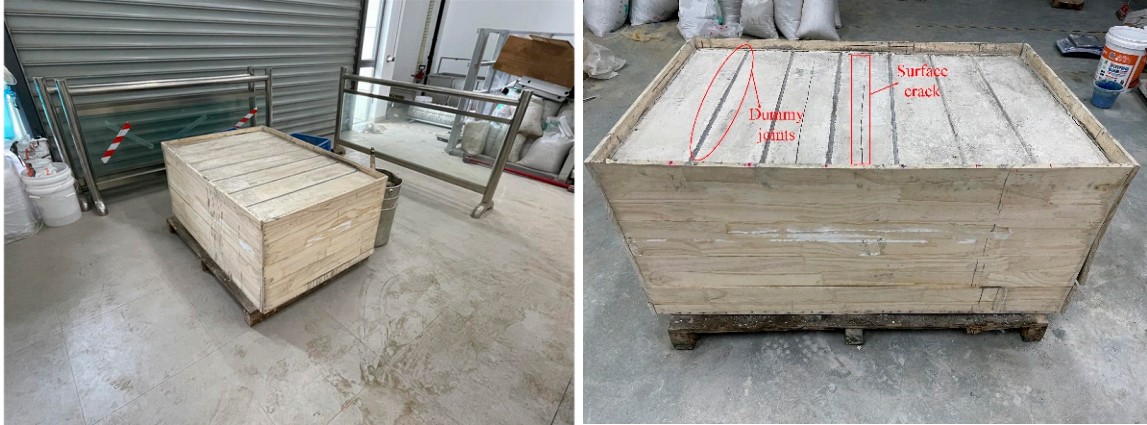

**Figure 12.** Scale model of slab track.

The temperature-control equipment used in the experiment was a temperature-controlled cabinet (type: UC27-60150-ES) with an internal size of 3 m × 3 m × 3 m, as shown in Figure 13. It can withstand a maximum load of 600 kg/m$^2$ and can provide a temperature range of −60 to 150 °C. Furthermore, it can allow the setting of the temperature–time curve and temperature-change rate, whereby rapid temperature rises and drops can be simulated.

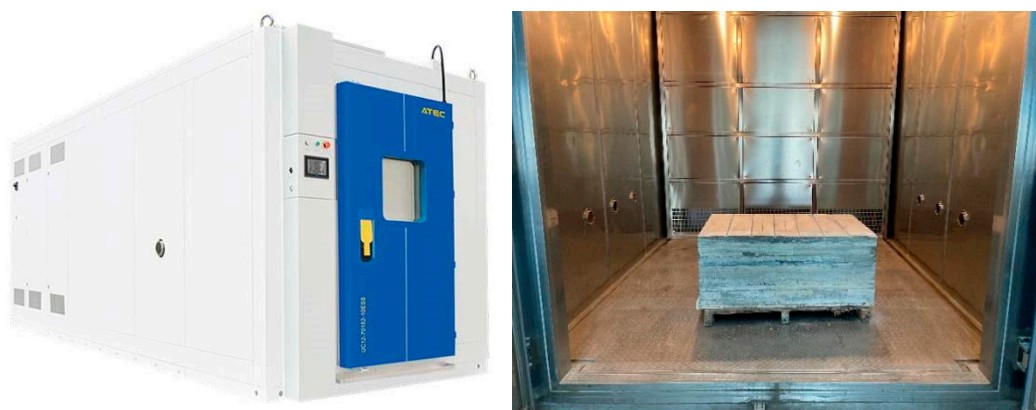

**Figure 13.** Slab-track-structure model in the temperature-controlled cabinet.

(3)    Temperature-change process and infrared thermogram

The slab-track-structure model was placed in the center of the bottom of the test box when heated, as shown in the right panel of Figure 13. The initial internal temperature was set to 20 °C, and the temperature-growth rate was set to 0.5 °C/min. After the temperature rose to 60 °C, it began to drop at a rate of 1 °C/min. During this process, the infrared thermogram was collected by a TIX620 thermal imager. The results of the temperature field on the slab surface are presented in Table 4 and Figure 14.

**Table 4.** The temperature in the crack and non-crack areas and DCJs on the slab surface.

| Time (min) | Internal Temp. (°C) | Temp. in Crack Areas (°C) | Temp. in Non-Crack Areas (°C) | Temp. Difference between Crack and Non-Crack Areas (°C) | Temp. at DCJs (°C) |
|---|---|---|---|---|---|
| 0 | 20 | 19.6 | 19.6 | 0 | 19.6 |
| 50 | 40 | 24.3 | 23.73 | 0.57 | 24.42 |
| 75 | 50 | 27.9 | 27.24 | 0.66 | 28.26 |
| 100 | 60 | 33.96 | 32.54 | 1.42 | 34.13 |
| 115 | 50 | 32.69 | 31.79 | 0.9 | 32.91 |
| 130 | 40 | 31.69 | 30.95 | 0.74 | 31.73 |
| 145 | 30 | 28.45 | 27.92 | 0.53 | 28.84 |
| 160 | 20 | 26.97 | 26.8 | 0.17 | 27.03 |
| 175 | 10 | 23.68 | 23.58 | 0.1 | 23.76 |

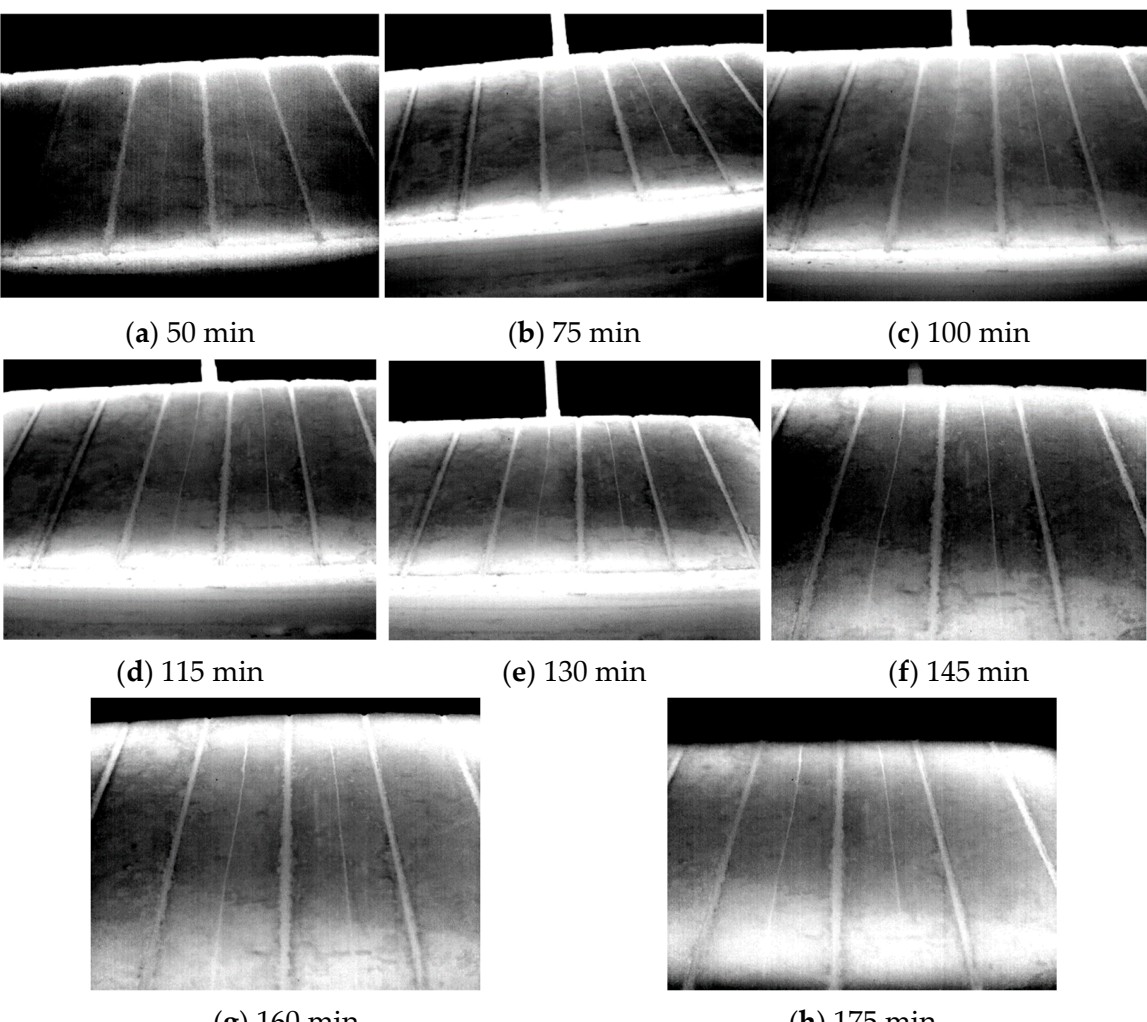

(**a**) 50 min  (**b**) 75 min  (**c**) 100 min

(**d**) 115 min  (**e**) 130 min  (**f**) 145 min

(**g**) 160 min  (**h**) 175 min

**Figure 14.** Infrared thermograms of slab surface of model track at different times.

*4.2. Results Analysis*

To extract the crack features, the thermograms in Figure 14 were processed by the proposed image-processing algorithm, and the results are shown in Figures 15 and 16. It can be observed that through the image enhancement by NSCT and the edge-detection algorithm based on phase congruency, two cracks on the slab surface were effectively identified.

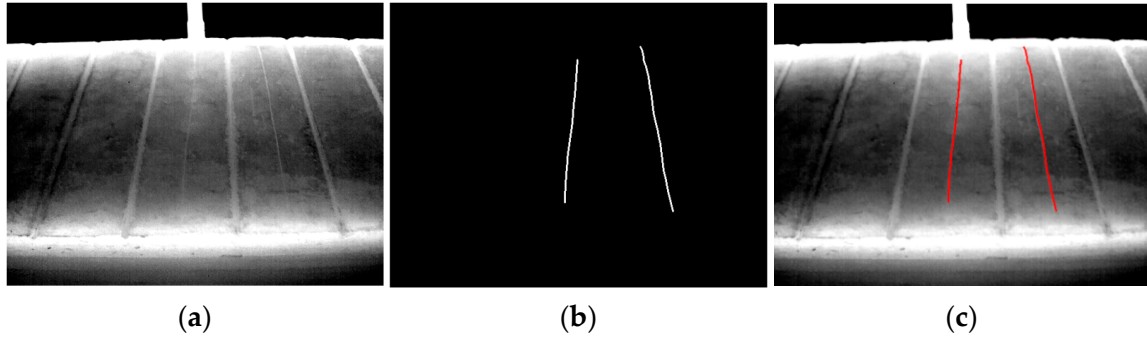

**(a)**         **(b)**         **(c)**

**Figure 15.** Infrared thermogram of track-slab surface at 100 min and crack-extraction results: (**a**) infrared thermogram; (**b**) extracted surface crack; (**c**) surface crack on infrared thermogram.

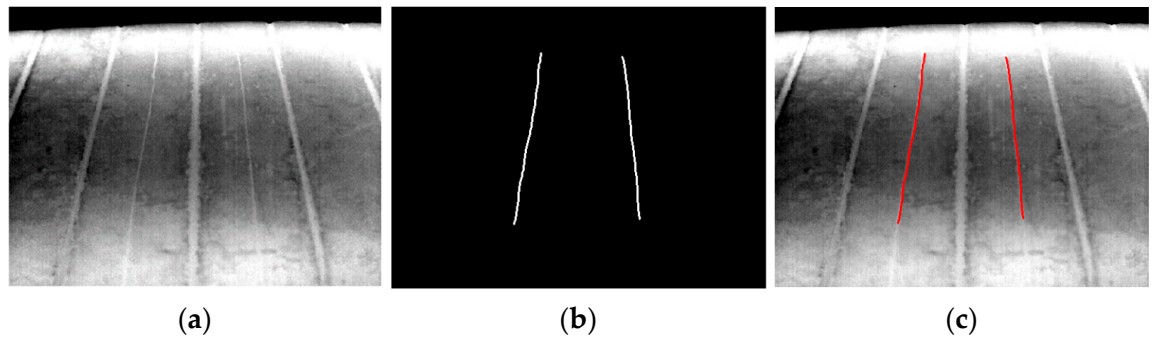

**(a)**         **(b)**         **(c)**

**Figure 16.** Infrared thermogram of track-slab surface at 130 min and crack-extraction results: (**a**) infrared thermogram; (**b**) extracted surface crack; (**c**) surface crack on infrared thermogram.

Next, the detailed information about the cracks, including the lengths, widths, and areas, was calculated using the proposed quantitative crack-detection method. Since the two cracks were prefabricated with the same length and width, this paper only presents the quantitative detection results of one of them, as listed in Table 5.

**Table 5.** Quantitative detection results of surface cracks at different times.

| Time/min | Width Detected/mm | Actual Width/mm | Acc. | Length Detected/mm | Actual Length/mm | Acc. | Area Detected/mm² | Actual Area/mm² | Acc. |
|---|---|---|---|---|---|---|---|---|---|
| 50 | 1.193 | | 99.4% | 689 | | 80.4% | 821.9 | | 80.1% |
| 75 | 1.184 | | 98.7% | 749 | | 87.4% | 886.8 | | 86.2% |
| 100 | 1.197 | | 99.8% | 722 | | 84.2% | 864.2 | | 84.0% |
| 115 | 1.198 | | 99.8% | 721 | | 84.1% | 863.8 | | 84.0% |
| 130 | 1.199 | 1.2 | 99.9% | 719 | 857 | 83.9% | 862.1 | 1028.4 | 83.8% |
| 145 | 1.196 | | 99.7% | 731 | | 85.3% | 874.3 | | 85.0% |
| 160 | 1.189 | | 99.1% | 720 | | 84.0% | 856.1 | | 83.2% |
| 175 | 1.191 | | 99.3% | 738 | | 86.1% | 878.9 | | 85.5% |

It can be observed from Table 5 that the detection accuracy of the proposed algorithm for the crack width was higher than 98.5%, but the accuracy in the detection of the length and area was only about 85%. The main reason for this problem is that in the infrared thermogram, the surface temperature at the edge of the track slab was higher than that in the middle area, which may have reduced the contrast between the crack and non-crack areas at the edge of the track slab. As a result, it was difficult to detect the cracks on the edge of the slab and the calculated length was shorter than the actual length of the cracks. Nevertheless, according to the current slab-track maintenance rules for HSRs, the level of surface cracking was only determined by its width, with no relevant information on length. Therefore, although the accuracy of the proposed method is limited, to some

degree, for predicting the lengths of slab-surface cracks, it can still meet the actual needs of HSR-slab-track maintenance due to its high performance in the prediction of crack width.

## 5. Concluding Remarks

In view of the actual condition of HSR maintenance within the relevant time window, infrared thermography can be an effective approach to detecting the surface cracks in track slabs. The aim of this paper was to solve the key problem of the quantitative detection of surface cracks based on the infrared thermogram of a slab track. The main contribution of this paper is the development of a thermographic processing method which can extract detailed information on surface cracks in slab tracks. It consists of two steps: feature extraction and quantitative detection. In the process of crack-feature extraction, this paper proposes a novel infrared-image-enhancement algorithm based on the NSCT multiscale product-thresholding method and a phase−congruency-based crack-edge-detection method. In the quantitative detection process, the lengths, widths, and areas of surface cracks are obtained based on the calculation of pixel numbers in the crack areas. Finally, the proposed method was verified by a laboratory test, in which a scale track model was used. The main conclusions are as follows:

(1) The proposed infrared-image enhancement algorithm can effectively solve the problem of fuzzy edge details and low contrast in the infrared thermograms of slab tracks and strengthen the crack-edge details in the image;

(2) The crack area can be located from the infrared thermogram via the edge-detection algorithm and morphological processing algorithm; the morphological processing method can be used to effectively remove the isolated noise and false edge areas in the image;

(3) The scale model of the track slab designed in this study can reflect the surface-temperature field of the original slab-track structure based on the results of the FE simulation of the slab-temperature field;

(4) As verified by the laboratory test, the quantitative accuracy of the detection of crack width by the proposed algorithm can be higher than 90%, whilst the accuracy of the algorithm's length and area detection is 85%, which essentially meets the requirements of current track maintenance for HSRs.

In summary, the proposed method can be used to improve the detection of surface cracks in HSR slab tracks based on thermal imaging, and its quantitative detection ability was verified. However, the actual operational condition of HSRs are much more complex than those in the laboratory. The combined effect of ambient temperature, rainwater, waterproof materials on the slab surface, and other external factors probably influence the accuracy with which surface cracks are detected. Future research may focus on how to further improve the effectiveness of IRT-based detection for slab tracks under the actual condition of HSRs. Furthermore, to improve the application of the proposed method in real HSR slab tracks, a track-inspection trolley equipped with an IR camera should be developed. However the parameters of the trolley, including the distance between the camera and the slab surface, the velocity of the trolley, and the frame rate, need further study to meet the requirements of crack inspection.

**Author Contributions:** Conceptualization, X.-Y.Y. and Y.-Y.L.; methodology, X.-Y.Y., Y.-Y.L. and Z.-W.L.; modeling, X.-Y.Y. and X.-Z.L.; validation, Z.-W.L. and X.-Z.L.; formal analysis, X.-Y.Y. and Y.-Y.L.; investigation, Z.-W.L. and X.-Z.L.; resources, Y.-Y.L. and X.-Z.L.; data curation, X.-Y.Y. and Z.-W.L.; writing—original draft preparation, X.-Y.Y. and X.-Z.L.; writing—review and editing, Y.-Y.L. and Z.-W.L.; visualization, X.-Z.L.; supervision, Y.-Y.L. and Z.-W.L.; project administration, Y.-Y.L.; funding acquisition, Y.-Y.L. and Z.-W.L. All authors have read and agreed to the published version of the manuscript.

**Funding:** This research was funded by the National Natural Science Foundation of China (grant nos. 52178430 and 52208441), Innovation Initiative Program of Shanghai (grant no. 20dz1203104), and Natural Science Foundation of Top Talent of SZTU (grant no. GDRC202128).

**Institutional Review Board Statement:** Not applicable.

**Informed Consent Statement:** Not applicable.

**Data Availability Statement:** All data, models, or code that support the findings of this study are available from the corresponding author upon reasonable request.

**Conflicts of Interest:** The authors declare no conflict of interest.

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
