# Peer review of "A Quantitative Detection Method for Surface Cracks on Slab Track Based on Infrared Thermography"

_applsci, doi:10.3390/app13116681_

Round 1

Reviewer 1 Report

A Quantitative Detection Method of Surface Crack on Slab Track Based on Infrared Thermography is an interesting article focused on important area of detection and quantification of cracks in high-speed rail slab tracks.

The article describes the theory of  IR images processing (transformation) to find the crack – I’m not able to assess its correctness because it is out of my field.

I appreciate that the article is enriched by an experiment. But I think that it should be more described in detail – there is only written camera type (no parameters like resolution of its detector, accuracy, sensitivity, etc. are presented). The emissivity used for temperature evaluation and the distance between camera and slab is not stated.

I don’t understand why do authors use the temperature control equipment? Do you try to prove the temperature independence of the used method or do you want to find optimal temperature for crack detection?

Do authors think about the real use of the IR camera on an inspection vehicle that can be used for cracks detection on the real railway? Are there any limits to its velocity in connection with the frequency of the IR camera and used optics (lens)?  

Comments

line 15

Then, to quantitatively detect the surface crack,  a pixel locating method

·         there is a big space between „crack,“ and „a pixel“ (maybe double space)

high-speed rail (HSR) – whole words and abbreviation - is repeated many times (lines 9, 22, 29, 35)

line 63

temtrack slabs with thermal imperature field – unknown words – needs to be explain

Reviewer 2 Report

The paper describes image processing methods to analyze infrared images for localizing surface cracks in slab track. In the second part of the paper results of a laboratory setup in a temperature controlled cabinet are presented.

After the introduction the first section presents an image enhancement algorithm (NSCT). But it is not described how the analyzed images are obtained. Is the IR camera recording the surface temperature over specific time duration? How long? Are temperature changes due to natural environment changes (sunshine, night?) How it is planned to be used for inspection of a real slab track? Please add details to the IR camera, e.g. how many images and with which frequency are recorded, what is the spatial resolution of the images?

The proposed image processing algorithms are described very detailed, but the description how the images are obtained is missing. Please extend the paper in a way that also the proposed experimental setup and the origin of the analyzed images are clear. This should be described before the image processing is discussed. Additionally, it should be also in a short way included in the abstract.  

Reviewer 3 Report

The paper shows promising image processing for crack detection on track slabs with infrared thermography. Nonetheless, the paper is challenging to read. As a reader, I would appreciate at least a block schema of the proposed method. It should also be highlighted what is new. Your result should be compared at least with one standard algorithm for crack detection. It is far from ideal to show results only on two different cracks. The article should suggest some limits, e.g., the size of crack which could be detected without any problem. 

Which programs/programming language did you use for the image processing and simulations?

Why do you think it is better to use an infrared camera than a visible camera? Maybe it would be worth to add that to the text.

How fast is your method, can it be used in real-time?

Could you explain a little bit in more detail how the technique should work in real life? The slab has almost 3 m and the resolution is 640 x 480. It leads to very low-resolution pixel/mm. I believe that means you can detect only large cracks. Is it enough?

Some additional notes:

[1,2] there is missing space

Line 47 etc.  Maybe it is better to leave it out as you write about the most used methods.

63 temtrack??

63 imperature???

Figure 9 y axis should bet the same for each image … 30-65°C

397 the camera is Fluke TIX620? There is missing plenty of information about the camera in your text which is crucial for interpretation of results… resolution, NETD...

Line 386 3m*3m*3m is usually not written this way

Table 3 usually it is written like a variable (unit) … Time (min)

Equations 14 and 15 are the same.

450 the quantitative detection accuracy of crack widths of the proposed algorithm can reach 99%. You showed only two cracks of the same size. This is misleading.

Round 2

Reviewer 2 Report

The new Table 1 is very useful. But it is a bit confusing that the two pairs of Category-Value are side by side, it would be better to have all the Category-Value pairs below each other.

IFOV =0.85 mRad does not define the spatial resolution, but the resolution in the FOV. The spatial resolution would be in mm/pixel, which is affected by many factors (optics, distance, number of pixels, ..). But this value is important, as if the authors want to detect defects with a width of 0.2-0.5 mm, then at least three pixels should be required for localizing a defect, which furthermore needs an appropriate spatial resolution. Please comment it.

Reviewer 3 Report

The authors have made significant changes. The article can be published as is.

I found minor things that should be corrected.

Figure 1. There is a typo – binery/binary

Line 85 Normally, the infrared thermography of the track slab can be obtained by an infrared camera. This is a strange formulation. Consider leaving it out.

Figure 2. b) is too close to the right edge.

Figure 11. It appears that the font is too small.
